# IN-CONTEXT PLANNING WITH LATENT TEMPORAL AB-STRACTIONS

## ABSTRACT

Planning-based reinforcement learning in real-world control faces two coupled obstacles: planning at primitive time scales explodes both context length and branching factor, and the underlying dynamics are often only partially observable. We introduce the *In-Context Latent Temporal Abstraction Planner* (*I-TAP*), which unifies in-context adaptation and online planning in a learned latent temporal-abstraction space. From offline trajectories, I-TAP learns an observation-conditioned residual-quantization VAE (RQ-VAE) that discretizes observation–macro-action sequences into a coarse-to-fine stack of residual tokens, together with a residual-quantized temporal Transformer that autoregressively predicts these tokens from recent observation and macro-action histories. This sequence model serves jointly as a context-conditioned prior over abstract actions and a latent-space dynamics model. At inference, I-TAP plans with Monte Carlo Tree Search directly in token space, leveraging short histories to implicitly infer latent factors without any test time fine-tuning. Across deterministic and stochastic MuJoCo locomotion and high-dimensional Adroit manipulation, including partially observable variants, I-TAP consistently matches or outperforms strong model-free and model-based baselines, demonstrating effective in-context planning under stochastic dynamics and partial observability.

## 1 INTRODUCTION

Since the introduction of Transformers (Vaswani et al., 2017), their versatility in handling diverse tasks has been widely recognized across domains (Brown et al., 2020; Bubeck et al., 2023). Recently, their sequence generation capability has been leveraged in planning-based reinforcement learning (RL), and subsequent work extends it to high-dimensional continuous control by learning temporal abstractions (e.g., options (Sutton et al., 1999), macro-actions (Mcgovern & Sutton, 1998)) and planning over high-level decisions (Jiang et al., 2023; Luo et al., 2025). However, these approaches often struggle when environments are partially observable and governed by latent, slowly varying parameters (e.g., wind disturbances for unmanned aerial vehicles or payload shifts for manipulators) that vary across scenarios.

In parallel, training Transformers on collected experience with sequence objectives (Chen et al., 2021) has enabled in-context adaptation, allowing models to adapt or self-improve at inference time without gradient updates (Brown et al., 2020; Furuta et al., 2022; Liu & Abbeel, 2023; Laskin et al., 2023; Huang et al., 2024). Yet most in-context RL methods either (i) lack an integrated planner at inference, thereby inheriting suboptimal behavior from the source algorithms they imitate (Son et al., 2025) and failing to convert predictions into optimized decisions under uncertainty (Paster et al., 2022) or (ii) operate at the primitive action level, which potentially scales poorly to high-dimensional continuous action spaces.

Motivated by these challenges, we introduce an in-context planning framework in learned latent temporal abstraction space. Our premise is that integrating temporal abstraction with in-context adaptation for planning-based RL addresses these issues jointly, analogous to how humans leverage high level concept to plan at multiple temporal scales. Adapting from recent histories and planning conditioned on them in a latent temporal abstraction space allows an agent to: (i) decouple adaptation and planning from the native temporal granularity of the Markov decision process (MDP), thereby shortening required context, easing the learning of a strong sequence model prior, and reducing the

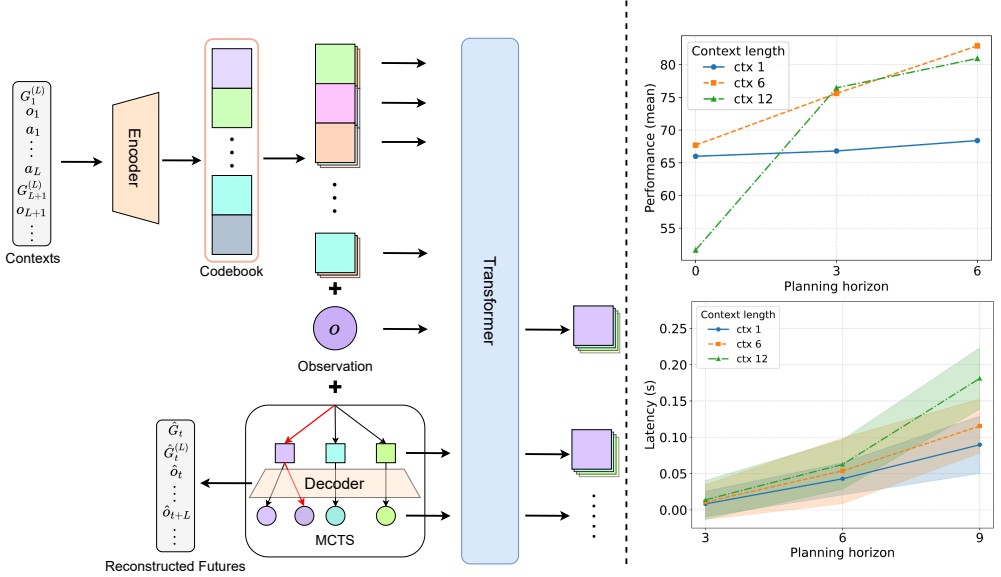

Figure 1: Overview of I-TAP. **Left:** A residual-quantized VAE (RQ-VAE) discretizes continuous state–macro-action trajectories into a coarse-to-fine token stack. **Right:** Normalized return and per-decision latency as functions of planning horizon and context size on Stochastic MuJoCo, highlighting the importance of a properly sized context window for effective in-context planning under stochasticity and partial observability.

branching factor during planning; (ii) use context to infer global latent parameters that govern the environment's dynamics (e.g., unobserved perturbation forces), enabling effective adaptation and forecasting across scenario shifts; and (iii) employ an online planner such as Monte Carlo Tree Search (MCTS) for exploration and optimization, providing a mechanism to deviate from suboptimal source policies and to handle uncertainty.

To this end, we propose *In-Context Latent Temporal Abstraction Planner* (I-TAP), which learns an in-context temporal abstraction model from offline data and performs online planning with the learned models to enable adaptive exploration and control. To extend the scalability and flexibility of our framework to handle high-dimensional temporal abstraction spaces, as illustrated in Fig. 1, we use an observation-conditioned residual quantized Variational AutoEncoder (RQ-VAE) (Lee et al., 2022) and train a residual-quantized temporal transformer (RQ-TT) (Lee et al., 2022) that autoregressively predicts a coarse-to-fine stack of residual tokens per time step. The RQ-TT thereby functions both as an in-context action-selection prior and as a dynamics estimator. For planning efficiency, inspired by Luo et al. (2025), I-TAP precomputes latent representations of plausible future trajectories conditioned on recent history in the latent space; Monte Carlo Tree Search (MCTS) then operates over these latent tokens using context-guided priors to balance exploration and exploitation under uncertainty; finally, we decode the selected latent stack to a primitive action sequence and execute the first action.

Our experiments reveal strong adaptability and performance of I-TAP. We train a single model on datasets collected under behavior policies of varying quality and across multiple hidden-parameter settings, and evaluate it in environments ranging from deterministic to highly stochastic. The same I-TAP model either outperforms or matches existing offline RL methods (Kumar et al., 2020; Kostrikov et al., 2022; Chen et al., 2021; Jiang et al., 2023; Rigter et al., 2023; Luo et al., 2025), demonstrating in-context adaptation, robust handling of environmental uncertainty, and the ability to deviate from suboptimal actions. We further show I-TAP's scalability to high-dimensional continuous action spaces in both fully and partially observed settings. Overall, across all tested conditions, I-TAP consistently adapts under partial observability and changing dynamics, scales to high-dimensional continuous control, and achieves planning efficiency through latent temporal abstraction.

## 2 BACKGROUND

**HiP-POMDP with Macro-Actions**   We consider the learning problem within the context of a Hidden-Parameter Partially Observable Markov Decision Process (HiP-POMDP) (Shaj et al., 2022; Doshi-Velez & Konidaris, 2016; Killian et al., 2017) represented by the tuple $(\mathcal{S}, \mathcal{O}, \mathcal{A}, \mathcal{W}, P, \mathcal{R})$, where $\mathcal{S} \subseteq \mathbb{R}^n$ is the latent (unobserved) state space, $\mathcal{O} \subseteq \mathbb{R}^d$ is the observation space, $\mathcal{A} \subseteq \mathbb{R}^\ell$ is the primitive-action space, and $\mathcal{W}$ is the space of hidden parameters. Each episode is associated with a latent task parameter $w \in \mathcal{W}$, which remains fixed throughout an episode but varies between episodes, influencing the dynamics globally. Our setting is closely related to the canonical HiP-MDPs, with the distinction that the state is partially observable to the agent.

The state transitions depend explicitly on the hidden parameter $w$ as follows: $s_{t+1} \sim P\big(s_{t+1} \mid s_t, a_t, w\big)$, where $s_t, s_{t+1} \in \mathcal{S}$ denote the latent state. At each step $t$ the agent observes $o_t \in \mathcal{O}$, chooses a primitive action $a_t \in \mathcal{A}$, and receives a reward $r_t = \mathcal{R}(s_t, a_t)$ together with the next observation $o_{t+1}$.

To manage complexity in continuous spaces and long-horizon planning, inspired by Luo et al. (2025), we introduce a finite set of macro-actions $\mathcal{M} = \{m = \langle a_t, \ldots, a_{t+L-1} \rangle \mid a_i \in \mathcal{A}\}$ with fixed duration $L$. Executing a macro-action $m$ from latent state $s_t$, conditioned on hidden parameter $w$, induces an $L$-step transition distribution $\tilde{P}\big(s_{t+L} \mid s_t, m, w\big) = P\big(s_{t+L} \mid s_t, a_t, \ldots, a_{t+L-1}, w\big)$, and an expected cumulative discounted reward $\tilde{r}(s_t, m, w) = \mathbb{E}_{s_{t+1:t+L} \sim P}\Big[\sum_{i=0}^{L-1} \gamma^i \mathcal{R}(s_{t+i}, a_{t+i})\Big]$. We define macro-policies as a mapping from the agent's finite interaction history to the set of macro-actions. Let $\mathcal{G}^{(L)} \subseteq \mathbb{R}$ denote the range of $L$-step discounted returns and define the augmented observation space $\bar{\mathcal{O}} := \mathcal{O} \times \mathcal{G}^{(L)}$. At macro boundaries $t_b = (b-1)L + 1$, we form the augmented observation $\bar{o}_{t_b} := \big(o_{t_b}, G_{t_{b-1}}^{(L)}\big)$ with the convention $G_{t_0}^{(L)} = 0$. Accordingly, we set $\mathcal{H} = \bigcup_{t \geq 0} (\bar{\mathcal{O}} \times \mathcal{A})^t \times \bar{\mathcal{O}}$, and let $h_t \in \mathcal{H}$ denote the history available at step $t$. The agent's decision-making is guided by a policy $\pi : \mathcal{H} \to \Delta(\mathcal{M})$. The objective is to find the optimal macro-policy $\pi^*$, such that $\pi^* = \arg\max_\pi \; \mathbb{E}_{w \sim p(w), \, \pi}\Big[\sum_{b=0}^{\infty} \gamma^{bL} \, \tilde{r}(s_{bL}, m_b, w)\Big]$.

**Trajectory Representation**   Consider an episode with $T = NL$ primitive steps, partitioned into $N$ macro-steps of fixed length $L$; let $t_b = (b-1)L + 1$ denote the start of macro-step $b$. The return-to-go from time $t$ is $G_t = \sum_{i=t}^{T} \gamma^{i-t} r_i$, and the $L$-step discounted return is $G_t^{(L)} = \sum_{j=0}^{L-1} \gamma^j r_{t+j}$. We write the macro-level trajectory as $\tau = ((G_{t_b}, G_{t_b}^{(L)}, o_{t_b}, m_{t_b}))_{b=1}^{N}$, where $o_{t_b}$ is the observation at $t_b$ and $m_{t_b}$ is the macro-action executed for $L$ primitive steps. This representation preserves the original dynamics while exposing temporally extended actions, forming the foundation for our offline reinforcement learning approach.

## 3 METHOD

In this section, we provide an in-depth explanation of each component of I-TAP: the discretization of state-macro-actions sequences, modeling the in-context prior and transition distribution over latent codes, and the planning process with MCTS. In general, I-TAP is a generative model for trajectories conditioned on both current observation and recent historical context, allowing for efficient in-context planning and decision-making.

### 3.1 IN-CONTEXT RESIDUAL DISCRETIZATION OF STATE MACRO ACTION SEQUENCES

In-context RL approaches tend to replicate the suboptimal behaviors of the source algorithm (Son et al., 2025), necessitating the integration of a planning mechanism to deviate from suboptimal decisions. Meanwhile, a discrete action space simplifies the representation of action distributions and facilitates the use of advanced planning algorithms (Silver et al., 2018). To leverage these advantages, prior work has proposed state-conditioned Vector Quantized Variational Autoencoders (VQ-VAE) (Jiang et al., 2023) to discretize the state-action spaces and make latent action space compact. However, applying vector quantization directly to these feature vectors can lead to low reconstruction accuracy if their dimensionality is very high (Jiang et al., 2024). Given these insights

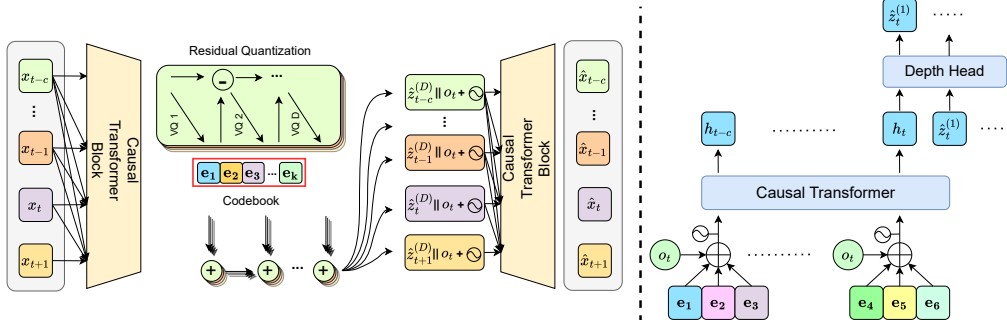

Figure 2: An overview of our RQ-VAE model that discretizes state-macro action sequences and temporal prior for I-TAP

and challenges, we use an observation-conditioned RQ-VAE to learn a discrete observation-macro action space to improve scalability.

**Tokens and masking.** At macro index $b$, a token is $x_{t_b} = \left(G_{t_b},\, G_{t_b}^{(L)},\, o_{t_b},\, m_{t_b}\right)$. For a training chunk $(x_{t_{b-c}}, \ldots, x_{t_b}, x_{t_{b+1}})$ with context length $c$. Inspired by Luo et al. (2025), the encoder masks the return-to-go $G_t$ at every position and includes the $L$-step return $G_t^{(L)}$ at all positions except the last two tokens $(x_{t_b}, x_{t_{b+1}})$, where the $G^{(L)}$ is masked. Such strategy provides a stable short-horizon signal throughout history while preventing reliance on these explicit returns due to their susceptibility to luck-induced variability in a stochastic environment (Paster et al., 2022).

**Residual Quantized VAE** A causal Transformer maps the chunk to per-token features $Z = \left(z_{t_{b-c}}, \ldots, z_{t_b}, z_{t_{b+1}}\right)$. We use a single codebook $E = \{e_1, \ldots, e_K\} \subset \mathbb{R}^d$ at all residual depths $d = 1, \ldots, D$ as using a shared codebook across depths yields a coarse-to-fine approximation with effective capacity up to $K^D$ without enlarging $K$ (Lee et al., 2022). For each time $t$,

$$r_t^{(0)} := z_t, \quad k_{t,d} = \arg\min_{k \in [K]} \left\| r_t^{(d-1)} - e_k \right\|_2^2, \quad r_t^{(d)} = r_t^{(d-1)} - e_{k_{t,d}},$$

and we define the depth-$d$ partial sum $\hat{z}_t^{(d)} := \sum_{j=1}^d e_{k_{t,j}}$. Assignments use a straight-through estimator; the codebook is updated by exponential moving average. Motivated by evidence from prior work (Jiang et al., 2023; Luo et al., 2023) that conditioning on state enables compact codebooks without sacrificing granularity. For each token, our decoder conditions on $o_{t_b}$ and the quantized latent $\hat{z}_t^{(D)}$ via a linear adapter and causal attention, which reconstruct all features for every token in the chunk leveraging its previous context: $(\hat{G}_{t_{b-c}}, \hat{G}_{t_{b-c}}^{(L)}, \hat{o}_{t_{b-c}}, \hat{m}_{t_{b-c}}, \ldots, \hat{G}_{t_{b+1}}, \hat{G}_{t_{b+1}}^{(L)}, \hat{o}_{t_{b+1}}, \hat{m}_{t_{b+1}})$. We optimize a reconstruction loss plus a residual partial-sum commitment:

$$\mathcal{L} = \sum_\tau \alpha_\tau \left\| (\hat{x}_\tau - x_\tau) \right\|_2^2 + \frac{\beta_{\mathrm{ps}}}{D} \sum_{d=1}^D \left\| Z - \mathrm{sg}\left[ \hat{Z}^{(d)} \right] \right\|_2^2.$$

Here $\alpha_\tau = \alpha_{\mathrm{tail}}$ for the last two tokens $(x_{t_b}, x_{t_{b+1}})$ and $\alpha_{\mathrm{ctx}}$ otherwise; $\hat{Z}^{(d)}$ is the depth-wise $d$ partial sum of residual code embeddings; and $\mathrm{sg}[\cdot]$ denotes stop-gradient. The depth-wise partial-sum term stabilizes residual quantization and prevents code hopping across depths, consistent with residual-quantized VAE (Lee et al., 2022).

### 3.2 Temporal Prior over Residual Code Stacks

Let $k_{t,1:D}$ denote the depth-$1{:}D$ codes at macro time $t$. We learn a depth-aware autoregressive prior that factorizes across time and within-time depth:

$$p_\phi\left(k_{t,1:D} \,\big|\, k_{<t,1:D},\, k_{t,<d},\, o_t\right) = \prod_{d=1}^D p_\phi\left(k_{t,d} \,\big|\, k_{<t,1:D},\, k_{t,<d},\, o_t\right).$$

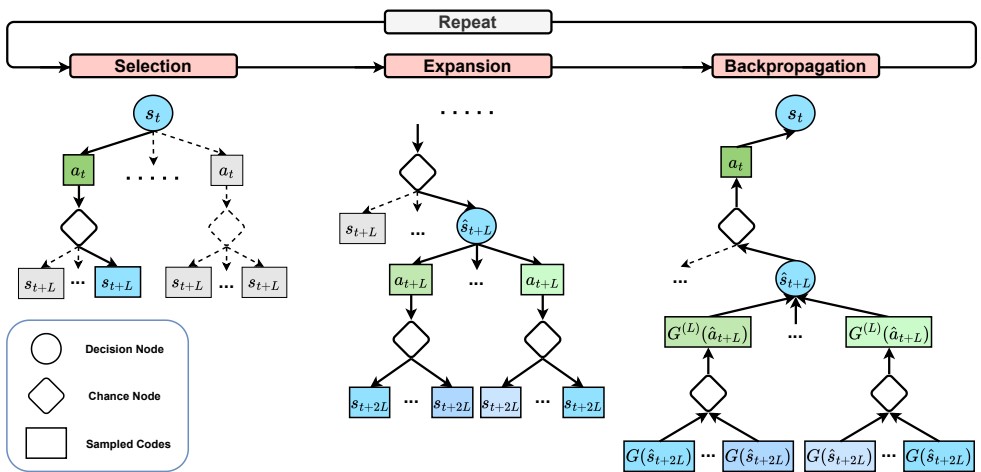

Figure 3: Macro-level MCTS overview. Each iteration uses P-UCT to select a macro-action, expands several candidates and their predicted outcomes in parallel, then backs up the resulting Q-estimates through the search tree to steer subsequent exploration.

A spatial trunk (causal over $t$) embeds each past position by the sum of its depth embeddings $\sum_{j=1}^{D} e_{k_{u,j}}$, adds positional and observation embeddings, and produces a context $h_t$. A lightweight *depth head* then predicts $k_{t,1}, k_{t,2}, \ldots, k_{t,D}$ by conditioning on $h_t$, a depth embedding, and the partial sum of shallower depths at time $t$. We then minimize the negative log-likelihood loss:

$$\mathcal{L}_{\text{prior}} \;=\; \mathbb{E}\Big[-\sum_t \sum_{d=1}^{D} \log p_\phi\big(k_{t,d} \,\big|\, k_{<t,1:D},\, k_{t,<d},\, o_t\big)\Big].$$

The time and depth factorization retains long temporal context while modeling within-position coarse to fine refinement efficiently.

### 3.3 In-Context Planning With Monte Carlo Tree Search

Prior work has leveraged MCTS to mitigate stochasticity arising from the environment in both online (Antonoglou et al., 2022) and offline RL (Luo et al., 2025). Planning in the real world, however, poses two additional challenges. First, partial observability induces apparent stochasticity when the context cannot reliably disambiguate latent states. Second, one needs a mechanism to balance the inherited bias and exploration at decision time when policies are distilled from suboptimal behavior.

We therefore adopt MCTS as the online planner in our latent temporal-abstraction space (Fig. 3.3). By taking expectations over futures under the learned latent dynamics, MCTS decouples action selection from noisy return estimates, explicitly handles uncertainty induced by partial observability, and provides a principled way to override suboptimal priors when predicted returns justify it. Furthermore, our planner couples P-UCT (Silver et al., 2017) with a context-conditioned prior over latent tokens and searches directly in the latent space, enabling targeted exploration while remaining in-distribution.

**Latent Decision Graph.** At time $t$, the agent observes $o_t$ and an interaction history in a sliding window of length $L \times c$. We encode this history into residual-quantized codes to obtain a context window $k_{t-1:t-c,1:D}$. A *decision node* is $s = (o_t, k_{t-1:t-c,1:D})$, and an *action edge* $a$ out of $s$ is a depth $D$ code stack $k_{t,1:D}$. Executing $a$ produces a distribution over *outcome codes* $k_{t+1,1:D}$ via our temporal prior $p_\phi(k_{t+1,1:D} \mid k_{t,1:D}, o_t, k_{t-1:t-c,1:D})$, and each outcome is decoded to a tail $(\hat{G}_{t+L}, \hat{G}_{t+L}^{(L)}, \hat{o}_{t+L}, \hat{m}_{t+L})$, yielding the successor decision node $s_{t+L}$. To mitigate the cost of iterative model calls with context (a bottleneck when parallelism is underused), motivated by Luo et al. (2025), we also prebuild a context-conditioned search space of plausible future latent code stacks, and run MCTS over this precomputed search space. The pseudocode of the process is shown in the section A.1

**Policy-guided selection**    From node $s$, we restrict top-$K$ action candidates to an action set $\mathcal{A}(s)$ and use a behavior-like prior to prioritize searching in-distribution actions without sacrificing exploration. Let the policy head produce probability logits $l_a$, we select $a$ by the AlphaZero-style P-UCT score:

$$a = \arg\max_a \Big[ Q(s,a) \ + \ \underbrace{\Big(c_1 + \log\frac{N(s)+c_2+1}{c_2}\Big)\frac{\sqrt{N(s)}}{1+N(s,a)}}_{\text{exploration term}} \ \underbrace{\Big(\pi_T(a\mid s) = \frac{e^{l_a/T}}{\sum_{b\in\mathcal{A}(s)} e^{l_b/T}}\Big)}_{\text{temperature scaled prior}} \Big],$$

where $N(s)$ and $N(s,a)$ are visit counts and $Q(s,a)$ is the action value; $c_1, c_2 > 0$ are P-UCT exploration constants. This coupling of a learned prior with P-UCT emphasizes in-distribution actions (large $\pi_T$) without sacrificing exploration via the exploration term, allowing the search to deviate from the source policy when returns warrant it.

**Parallel expansion and backpropagation.**    At each decision node $s_t$ we expand the top-$K$ candidates in parallel, sample $D$ outcome codes at the chance node, and decode them to obtain a successor $s_{t+L} = (\hat{o}_{t+L},\, k'_{t:t-c+1,\, 1:D})$ and its leaf value. We then back up $Q(s,a)$ along the visited path using incremental averages and update visit counts.

## 4    EXPERIMENTS

We evaluate I-TAP through comprehensive empirical studies using tasks from the D4RL benchmark (Fu et al., 2020), focusing on standard Gym locomotion tasks and complex high-dimensional Adroit robotic manipulation. Our experiments assess both performance and adaptability of I-TAP across varying degrees of environmental stochasticity and partial observability. We additionally conduct ablation studies to examine how macro-action length, context length, planning horizon, and residual depth affect performance. We further analyze the relationship between decision latency and context length in Appendix A.2.

**Baselines.**    We compare I-TAP to strong offline RL baselines: model-free actor–critic methods Conservative Q-Learning (CQL; Kumar et al. (2020)) and Implicit Q-Learning (IQL; Kostrikov et al. (2022)); context-conditioned policy methods such as Decision Transformer (DT; Chen et al. (2021)); and model-based planners that operate over learned temporal abstractions, including Trajectory Autoencoding Planner (TAP; Jiang et al. (2023)), Latent Macro-Action Planner (L-MAP; Luo et al. (2025)), where TAP is largely insensitive to raw action dimensionality and shows strong performance on high-dimensional Adroit manipulation, and L-MAP likewise scales well to high-dimensional control while remaining robust under stochastic dynamics. Finally, a risk-sensitive, model-based specialist for stochastic domains (1R2R; Rigter et al. (2023)).

**Experimental Setup.**    To assess I-TAP's adaptation capabilities across varying latent task parameters and associated dynamics, we conduct comprehensive experiments using the Stochastic MuJoCo tasks introduced by Rigter et al. (2023). Each environment defines a global latent task parameter controlling perturbation levels (Deterministic, Moderate-Noise, High-Noise), which in turn specify the distribution of instantaneous hidden forces at each step. We train a single model per method (I-TAP, DT, L-MAP) on the union of datasets stratified by behavior-policy quality, and dataset-specific models for the remaining baselines. To test scalability to high-dimensional control and partial observability, we evaluate on Adroit (Rajeswaran et al., 2018) in (i) the original fully observable setting and (ii) a partially observable variant where we mask a subset of target-position coordinates. Unless otherwise noted, we set the latent context size to $C = 6$ tokens; with macro length $L = 3$, this window summarizes 18 past primitive transitions. Further domain and hyperparameter details appear in Appendix A.3.

### 4.1    MAIN RESULTS

**Mujoco**    We use MuJoCo tasks to evaluate I-TAP's in-context adaptation and its robustness to uncertainty under partial observability. Table 1 reports normalized scores across noise regimes and dataset qualities. I-TAP attains the highest mean in every dataset type and demonstrates strong adaptability across varying environment dynamics without gradient update. These gains reflect I-TAP's ability to (i) leverage history for in-context adaptation and (ii) plan online to optimize decisions rather than simply imitating the dataset policy. When compared to L-MAP, I-TAP's better

Table 1: Normalised results for high-noise (High), moderate-noise (Mod), deterministic ($-$) environments. Bold numbers indicate the best scores in each row.

| Dataset Type | Env (Noise) | Model-Based | | | | Model-Free | | |
|---|---|---|---|---|---|---|---|---|
| | | I-TAP | L-MAP | TAP | 1R2R | DT | CQL | IQL |
| Medium-Expert | Hopper (High) | **82.87 ± 3.55** | 71.49 ± 3.46 | 37.31 ± 3.66 | 37.99 ± 2.71 | 61.87 ± 2.56 | 68.03 ± 3.94 | 44.83 ± 2.58 |
| | Hopper (Mod) | 104.45 ± 2.66 | 93.40 ± 3.65 | 40.86 ± 5.42 | 52.19 ± 8.37 | 73.86 ± 2.68 | **106.17 ± 2.16** | 60.61 ± 3.46 |
| | Hopper ($-$) | **111.71 ± 0.08** | 106.74 ± 2.24 | 85.55 ± 3.83 | 57.40 ± 6.06 | 101.6 ± 1.85 | 105.4 | 91.5 |
| | Walker2D (High) | **93.50 ± 3.15** | 92.75 ± 1.34 | 91.09 ± 2.78 | 32.38 ± 4.55 | 52.42 ± 1.27 | 83.18 ± 3.70 | 68.61 ± 3.33 |
| | Walker2D (Mod) | **97.29 ± 1.68** | 93.48 ± 1.20 | 91.40 ± 1.42 | 56.48 ± 7.51 | 64.67 ± 1.00 | 91.44 ± 1.44 | 86.66 ± 1.84 |
| | Walker2D ($-$) | 101.20 ± 1.91 | 100.38 ± 0.72 | 105.32 ± 2.03 | 73.18 ± 6.29 | 64.65 ± 0.79 | 108.8 | **109.6** |
| Mean (Medium-Expert) | | **98.50** | 93.04 | 75.26 | 51.60 | 69.85 | 93.84 | 76.97 |
| Medium | Hopper (High) | **67.80 ± 2.60** | 59.05 ± 2.93 | 43.93 ± 2.66 | 33.99 ± 0.92 | 55.91 ± 2.02 | 45.21 ± 2.97 | 49.69 ± 2.47 |
| | Hopper (Mod) | **72.47 ± 2.71** | 63.21 ± 3.10 | 43.64 ± 2.25 | 65.24 ± 3.31 | 60.97 ± 0.82 | 49.92 ± 3.00 | 56.00 ± 3.60 |
| | Hopper ($-$) | **81.94 ± 2.14** | 61.65 ± 2.81 | 69.14 ± 2.33 | 55.49 ± 3.99 | 58.14 ± 0.24 | 58.0 | 66.3 |
| | Walker2D (High) | 60.35 ± 2.73 | 59.05 ± 2.30 | 52.20 ± 2.76 | 32.13 ± 4.51 | 32.20 ± 0.83 | **61.49 ± 3.24** | 47.53 ± 3.05 |
| | Walker2D (Mod) | **65.85 ± 2.44** | 62.23 ± 1.84 | 44.46 ± 1.82 | 65.16 ± 2.84 | 43.77 ± 0.95 | 49.38 ± 2.02 | 48.82 ± 2.31 |
| | Walker2D ($-$) | **79.57 ± 1.22** | 75.54 ± 1.59 | 51.75 ± 3.30 | 55.69 ± 4.97 | 55.36 ± 0.61 | 72.5 | 78.3 |
| Mean (Medium) | | **71.33** | 63.46 | 50.85 | 51.28 | 51.06 | 56.08 | 57.77 |
| Medium-Replay | Hopper (High) | **70.67 ± 2.59** | 60.76 ± 2.79 | 48.69 ± 2.97 | 68.25 ± 3.78 | 35.17 ± 0.96 | 51.70 ± 3.09 | 43.27 ± 2.78 |
| | Hopper (Mod) | **81.33 ± 2.19** | 73.81 ± 2.67 | 38.10 ± 3.22 | 22.82 ± 2.08 | 35.76 ± 1.01 | 40.53 ± 1.52 | 49.12 ± 3.38 |
| | Hopper ($-$) | 86.57 ± 2.03 | 90.8 ± 0.63 | 80.92 ± 3.79 | 89.67 ± 1.92 | 43.01 ± 1.36 | **95.0** | 94.7 |
| | Walker2D (High) | **70.60 ± 3.07** | 59.16 ± 2.92 | 55.15 ± 3.29 | 65.63 ± 3.41 | 37.22 ± 0.78 | 50.33 ± 3.88 | 45.13 ± 2.38 |
| | Walker2D (Mod) | **74.04 ± 1.94** | 69.20 ± 2.55 | 43.49 ± 2.27 | 52.23 ± 2.22 | 49.51 ± 0.81 | 40.24 ± 1.67 | 40.77 ± 2.72 |
| | Walker2D ($-$) | 75.49 ± 2.52 | 70.66 ± 1.78 | 72.32 ± 3.26 | **90.67 ± 1.98** | 48.44 ± 0.76 | 77.2 | 77.2 |
| Mean (Medium-Replay) | | **76.45** | 70.73 | 56.45 | 64.88 | 41.52 | 59.17 | 58.37 |

performance indicates the value of context-guided action selection during search, which balances exploration with a sequence-modeling prior rather than relying solely on MCTS for improving robustness of decisions. In contrast, DT degrades under stochastic dynamics and with lower-quality behavior data, which is consistent with its lack of an integrated planner and its reliance on return conditioning, which is susceptible to luck-induced variability (Paster et al., 2022). By conditioning only on recent rewards and using a downstream planner, I-TAP avoids inheriting suboptimal behavior from the behavior policy and can deviate when the planner finds more promising actions.

Table 2: Adroit robotic hand control results.

| Dataset Type | Env | I-TAP | L-MAP | TAP |
|---|---|---|---|---|
| Cloned | Pen | **85.44 ± 8.19** | 60.68 ± 7.88 | 46.44 ± 7.54 |
| Cloned | Hammer | **4.38 ± 1.28** | 2.43 ± 0.29 | 1.32 ± 0.12 |
| Cloned | Door | **14.17 ± 1.34** | 13.22 ± 1.34 | 13.45 ± 1.43 |
| Cloned | Relocate | 0.08 ± 0.02 | **0.15 ± 0.13** | -0.23 ± 0.01 |
| Expert | Pen | **133.81 ± 5.23** | 126.60 ± 5.60 | 127.40 ± 7.70 |
| Expert | Hammer | **128.37 ± 0.21** | 127.16 ± 0.29 | 127.60 ± 1.70 |
| Expert | Door | **105.98 ± 0.08** | 105.24 ± 0.10 | 104.80 ± 0.80 |
| Expert | Relocate | **109.85 ± 0.88** | 107.57 ± 0.76 | 106.21 ± 1.61 |
| Expert (POMDP) | Pen | **81.68 ± 9.83** | 69.84 ± 9.81 | 60.87 ± 9.55 |
| Expert (POMDP) | Hammer | **72.36 ± 8.48** | 59.21 ± 6.52 | 42.22 ± 12.92 |
| Expert (POMDP) | Door | **95.05 ± 3.12** | 89.35 ± 3.41 | 83.71 ± 4.22 |
| Expert (POMDP) | Relocate | **52.16 ± 4.46** | 37.36 ± 3.84 | 33.94 ± 3.50 |
| **Mean (Expert)** | | **119.50** | 116.64 | 116.50 |
| **Mean (Expert POMDP)** | | **75.31** | 63.94 | 55.19 |

**Adroit Control** Adroit poses high-dimensional state–action spaces and fine-grained control demands. Table 2 shows that I-TAP achieves the best performance in both fully observable and partially observable setting. On cloned datasets, I-TAP outperforms both L-MAP and TAP in three of four tasks, indicating that planning with context-guided prior in the latent abstraction space enables the agent to do more targeted exploration with promising returns while remaining in distribution. On expert datasets, I-TAP matches or exceeds baselines, and under partial observability (POMDP) it maintains the top scores across all four tasks, highlighting the benefit of context-conditioned planning when state aliasing induces apparent stochasticity and its ability to leverage feedback from the environment to provide more targeted exploration to improve decisions. We use residual depth D=2

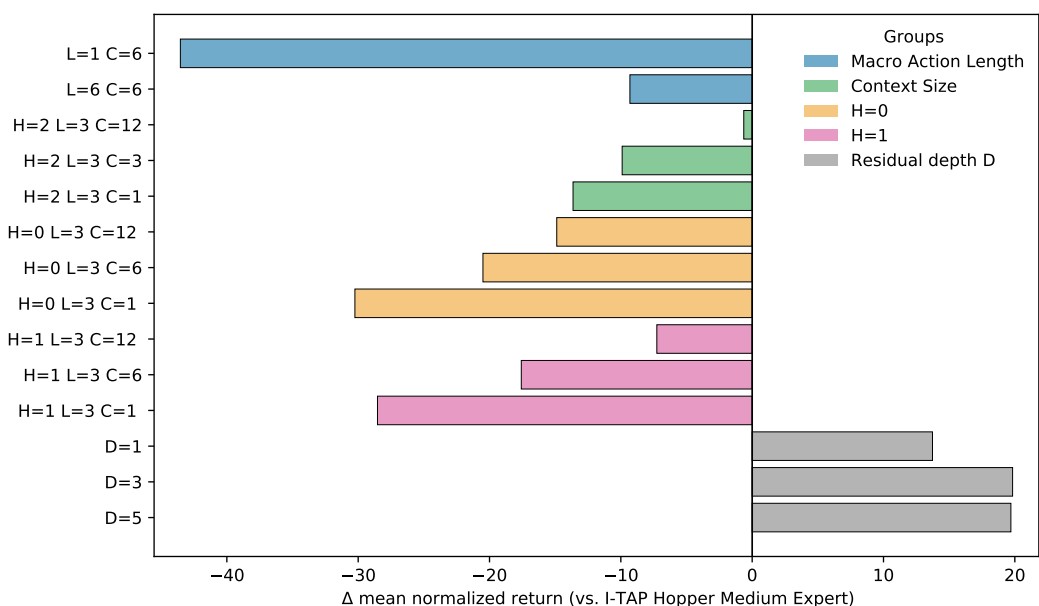

Figure 4: Ablation results across Adroit (expert) and MuJoCo Hopper. We plot $\Delta$ scores relative to I-TAP on Hopper medium–expert, which serves as the baseline (zero line).

for cloned and D=3 for expert datasets; the stronger expert results are consistent with the intuition that coarse-to-fine residual quantization helps retain the granularity needed for high-dimensional continuous control.

## 4.2 ABLATION STUDY

We present analyses and ablations of macro action length, context length, planning horizon, and residual depth. Figure 4 summarizes the results from ablation studies conducted across deterministic, moderate-noise, and high-noise Mujoco Hopper control tasks and Adroit (expert) tasks.

**Macro Action Length.** We vary the macro length $L$ and compare $L=1$ versus $L=6$. Increasing to $L=6$ does *not* noticeably degrade performance, whereas decreasing to $L=1$ causes a substantial drop. Two factors explain this: (i) with fixed $C$, the *effective history* observed by the planner scales as $C \times L$ steps, so $L=1$ shortens the usable history and weakens in-context adaptation; and (ii) shorter macros increase the *branching factor* during search and make the RQ-VAE more prone to *overfitting*, which requires careful hyperparameter control.

**Context Size.** We ablate context size $C \in \{1, 3, 12\}$ (in latent tokens). The datasets are produced under distinct latent dynamics; thus, context is useful for *inferring the active mode* and for feeding back recent rewards to steer exploration. Increasing context from $C=1$ (approximately 3 past transitions when $L=3$) to $C=12$ (approximately 36 transitions) yields consistent gains. *Takeaway:* short, *latent* contexts already help (few tokens cover many steps), and performance improves as $C$ grows because the model better disambiguates latent parameters and aggregates noisy feedback.

**Planning Horizon.** We vary the MCTS look-ahead depth $H$ (in latent tokens). With macro length $L=3$, a depth $H$ corresponds to $H \times L$ planning horizon in the raw action space. Removing planning ($H=0$) harms performance for all $C$; the drop is smaller on deterministic tasks where the *in-context prior* is already strong, but it remains insufficient under stochasticity. Increasing to $H=1$ yields consistent gains across settings; at $H=2$, we observe a pronounced jump, a shorter-context model ($C=6$) matches a longer-context baseline ($C=12$). This supports our hypothesis that MCTS could *mitigates uncertainty* due to partial observability and stochastic dynamics by taking expectations over futures: deeper look-ahead can substitute for additional context up to a point before compute and diminishing returns dominate.

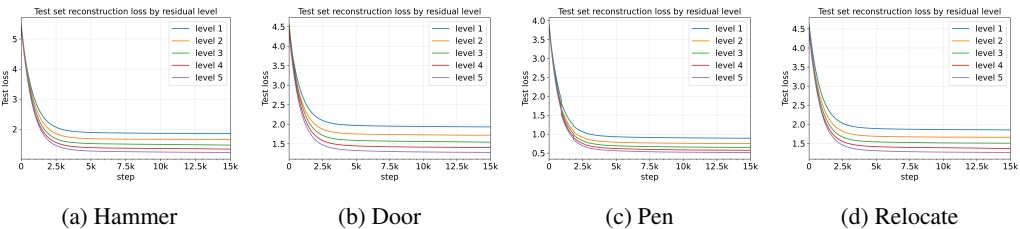

Figure 5: Test reconstruction losses across residual levels.

**Residual Depth** We vary the residual quantization depth $D$ for RQ-VAE. Reconstruction error *decreases monotonically* with $D$ (Fig. 4.2), with diminishing returns: the largest drop is from $D{=}1$ to $D{=}3$. Control performance mirrors this: the task-averaged score rises from 113.40 at $D{=}1$ to 119.50 at $D{=}3$, and then plateaus (119.37 at $D{=}5$). This shows that $D{=}3$ preserves action granularity for high-DoF control without incurring the compute/optimization overhead of very deep stacks.

## 5 RELATED WORKS

**Reinforcement Learning as Sequence Modeling.** Recent advances have reframed reinforcement learning (RL) as a sequence modeling problem, initiated by Decision Transformer (DT), which formulates RL as supervised learning conditioned on desired returns (Chen et al., 2021). Building on these ideas, Algorithm Distillation (Laskin et al., 2023) and Decision-Pretrained Transformer (Lee et al., 2023) leverage Transformers to distill optimal behaviors from historical trajectories, enabling rapid in-context adaptation. More recently, hierarchical variants like In-context Decision Transformer (IDT; Huang et al. (2024)) extend this paradigm by modeling high-level decisions, alleviating computational bottlenecks associated with long context windows. However, as noted by Son et al. (2025), these methods may replicate suboptimal behaviors due to the absence of explicit planning mechanisms, a limitation potentially addressed by model-based planning. Moreover, supervised RL methods typically assume deterministic or near-deterministic datasets, inherently limiting their effectiveness in stochastic environments, where conditioning solely on outcomes can lead to incorrect decisions (Paster et al., 2022). Unlike prior works relying on large contexts in near-deterministic settings, our approach explicitly targets efficient adaptation in dynamic, stochastic environments using a limited context window.

**Model-Based Reinforcement Learning.** From a model-based perspective (Antonoglou et al., 2022; Schrittwieser et al., 2020), Janner et al. (2021) introduced beam search over a Transformer dynamics model for planning, inspiring subsequent methods that plan in learned latent spaces. In particular, TAP (Jiang et al., 2023) and L-MAP (Luo et al., 2025) employ temporal abstraction by encoding multi-step action segments into discrete codes via state-conditioned VQ-VAEs, then planning over these compact tokens. While beam search with a learned model is effective in largely deterministic settings, L-MAP further adopts MCTS to handle stochastic dynamics and improve robustness. Nevertheless, both methods assume full observability, which can limit performance under state aliasing. Our proposed I-TAP bridges this gap by conditioning planning on recent histories to mitigate partial observability and use MCTS to take expectations over possible futures and deviate from suboptimal priors.

## 6 CONCLUSION

In this work, we introduced the *In-Context Latent Temporal Abstraction Planner* (I-TAP), a novel offline reinforcement learning approach for planning in partially observable continuous control tasks characterized by latent parameters and stochasticity arising from unobserved hidden variables. I-TAP integrates temporal abstraction into in-context planning to reduce planning complexity and adapt dynamically to variations across different scenarios, and employs MCTS to explicitly handle uncertainty. Extensive evaluations demonstrate that I-TAP consistently achieves superior or competitive performance relative to state-of-the-art baselines.

**Reproducibility statement.** We ensure reproducibility by providing our code in the supplementary material. We will publicly release the code upon the acceptance of this work.

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

# A  APPENDIX

## A.1  LATENT DECISION GRAPH CONSTRUCTION

---

**Algorithm 1**  SAMPLESTACK$(p_\phi, \mathcal{C}, D, \boldsymbol{\tau}, \boldsymbol{\rho})$

---

1: $\ell_1 \leftarrow$ logits from $p_\phi(k_1 \mid \mathcal{C})$
2: $k_1 \sim$ TOPKTEMPCAT$(\ell_1, \tau_1, \rho_1)$
3: $K \leftarrow (k_1)$
4: **for** $d = 2$ **to** $D$ **do**
5: $\quad \ell_d \leftarrow$ logits from $p_\phi(k_d \mid K_{1:d-1}, \mathcal{C})$
6: $\quad k_d \sim$ TOPKTEMPCAT$(\ell_d, \tau_d, \rho_d)$
7: $\quad K \leftarrow (K, k_d)$             ▷ append
8: **end for**
9: **return** $K$

---

**Algorithm 2**  PRE-CONSTRUCTING THE LATENT SEARCH SPACE (RESIDUAL STACK, MACRO TOKENS)

---

**Require:** Current observation $o_{t_k}$; context $\big((G_{t-cL}^{(L)}, o_{t-cL}, a_{t-cL}), \ldots, (G_{t-1}^{(L)}, o_{t-1}, a_{t-1})\big)$; encoder $f_{\text{enc}}$; decoder $f_{\text{dec}}$; residual-stack model $p_\phi$; residual depth $D$; per-depth temperatures $\boldsymbol{\tau} = (\tau_1, \ldots, \tau_D)$; per-depth top truncation $\boldsymbol{\rho} = (\rho_1, \ldots, \rho_D)$; # coarse samples $M$; # residual completions per coarse sample $J$; # lookahead samples $N$; tree depth $H$; # kept per node $\kappa_{\text{keep}}$; # proposals per node $B$

**Ensure:** Latent search tree $\mathcal{T}$ with cached promising residual-stack codes

1: **Encode macro-context**
2: $k_{t_{k-1}:t_{k-c}, 1:D} \leftarrow f_{\text{enc}}\big((G_{t-cL}^{(L)}, o_{t-cL}, a_{t-cL}), \ldots, (G_{t-1}^{(L)}, o_{t-1}, a_{t-1})\big)$
3: Initialize tree $\mathcal{T}$ with root node $s_{t_k} = (o_{t_k}, k_{t_{k-1}:t_{k-c}, 1:D})$
4: $\mathcal{C}_k \leftarrow (o_{t_k}, k_{t_{k-1}:t_{k-c}, 1:D})$
5: **Step 1: sample and score initial macro stacks at index** $k$
6: /* $M$ coarse draws for depth 1; for each, $J$ residual completions to depth $D$ */
7: $\ell_{k,1} \leftarrow$ logits from $p_\phi(k_{t_k,1} \mid \mathcal{C}_k)$
8: **for** $i = 1$ **to** $M$ **(parallel) do**
9: $\quad k_{t_k,1}^{(i)} \sim$ TOPKTEMPCAT$(\ell_{k,1}, \tau_1, \rho_1)$
10: $\quad$ **for** $j = 1$ **to** $J$ **(parallel) do**
11: $\quad\quad K_{t_k}^{(i,j)} \leftarrow (k_{t_k,1}^{(i)})$
12: $\quad\quad$ **for** $d = 2$ **to** $D$ **do**
13: $\quad\quad\quad \ell_{k,d} \leftarrow$ logits from $p_\phi\big(k_{t_k,d} \mid K_{t_k,1:d-1}^{(i,j)}, \mathcal{C}_k\big)$
14: $\quad\quad\quad k_{t_k,d}^{(i,j)} \sim$ TOPKTEMPCAT$(\ell_{k,d}, \tau_d, \rho_d)$
15: $\quad\quad\quad K_{t_k}^{(i,j)} \leftarrow (K_{t_k}^{(i,j)}, k_{t_k,d}^{(i,j)})$      ▷ append
16: $\quad\quad$ **end for**
17: $\quad\quad z_{t_k}^{(i,j)} \leftarrow$ Embed$(K_{t_k}^{(i,j)})$
18: $\quad\quad \widehat{G}_{t_k}^{(L)}(K_{t_k}^{(i,j)}) \leftarrow$ current-step head from $f_{\text{dec}}$
19: $\quad\quad$ **for** $n = 1$ **to** $N$ **(parallel) do**
20: $\quad\quad\quad \mathcal{C}' \leftarrow (o_{t_k}, K_{t_k}^{(i,j)}, k_{t_{k-1}:t_{k-c}, 1:D})$
21: $\quad\quad\quad K_{t_{k+1}}^{(i,j,n)} \leftarrow$ SampleStack$(p_\phi, \mathcal{C}', D, \boldsymbol{\tau}, \boldsymbol{\rho})$

22:             $z_{t_{k+1}}^{(i,j,n)} \leftarrow \text{Embed}(K_{t_{k+1}}^{(i,j,n)})$

23:             $\hat{y}_{t_{k+1}}^{(i,j,n)} \leftarrow f_{\text{dec}}(z_{t_k}^{(i,j)}, z_{t_{k+1}}^{(i,j,n)}, o_{t_k}, k_{t_{k-1}:t_{k-c}, 1:D})$

24:        **end for**

25:        $\text{score}(K_{t_k}^{(i,j)}) \leftarrow \frac{1}{N}\sum_{n=1}^{N}\left(\widehat{G}_{t_k}^{(L)}(K_{t_k}^{(i,j)}) + [\hat{y}_{t_{k+1}}^{(i,j,n)}]_{\text{rtg}}\right)$

26:        $\bar{y}_{t_{k+1}}^{(i,j)} \leftarrow \frac{1}{N}\sum_{n=1}^{N}\hat{y}_{t_{k+1}}^{(i,j,n)}; \quad \hat{o}_{t_{k+1}}^{(i,j)} \leftarrow \text{Obs}(\bar{y}_{t_{k+1}}^{(i,j)})$

27:      **end for**

28: **end for**

29: Select top-$\kappa_{\text{keep}}$ stacks $\{K_{t_k}^{(i,j)}\}$ by score; for each, attach child node $(\hat{o}_{t_{k+1}}^{(i,j)}, K_{t_k}^{(i,j)})$ under the root in $\mathcal{T}$

30: **Step 2: recursive latent-tree expansion over macro indices**

31: **for** $h = 2$ **to** $H$ **do**

32:      Let $\mathcal{N}_{h-1}$ be the nodes at depth $h-1$ of $\mathcal{T}$

33:      **for each** node $(\hat{o}, K^c) \in \mathcal{N}_{h-1}$ **(parallel) do**

34:        $\mathcal{C} \leftarrow (\hat{o}, K^c, k_{t_{k-1}:t_{k-c}, 1:D})$

35:        **for** $b = 1$ **to** $B$ **(parallel) do**

36:          $K_{t_{k+h-1}}^{(b)} \leftarrow \text{SampleStack}(p_\phi, \mathcal{C}, D, \boldsymbol{\tau}, \boldsymbol{\rho})$

37:          $z_{t_{k+h-1}}^{(b)} \leftarrow \text{Embed}(K_{t_{k+h-1}}^{(b)})$

38:          $\widehat{G}_{t_{k+h-1}}^{(L)}(K_{t_{k+h-1}}^{(b)}) \leftarrow$ current-step head from $f_{\text{dec}}$

39:          **for** $n = 1$ **to** $N$ **(parallel) do**

40:            $\mathcal{C}^\star \leftarrow (\hat{o}, K_{t_{k+h-1}}^{(b)}, k_{t_{k-1}:t_{k-c}, 1:D})$

41:            $K_{t_{k+h}}^{(b,n)} \leftarrow \text{SampleStack}(p_\phi, \mathcal{C}^\star, D, \boldsymbol{\tau}, \boldsymbol{\rho})$

42:            $z_{t_{k+h}}^{(b,n)} \leftarrow \text{Embed}(K_{t_{k+h}}^{(b,n)})$

43:            $\hat{y}_{t_{k+h}}^{(b,n)} \leftarrow f_{\text{dec}}(z_{t_{k+h-1}}^{(b)}, z_{t_{k+h}}^{(b,n)}, \hat{o}, k_{t_{k-1}:t_{k-c}, 1:D})$

44:          **end for**

45:          $\text{score}(K_{t_{k+h-1}}^{(b)}) \leftarrow \frac{1}{N}\sum_{n=1}^{N}\left(\widehat{G}_{t_{k+h-1}}^{(L)}(K_{t_{k+h-1}}^{(b)}) + [\hat{y}_{t_{k+h}}^{(b,n)}]_{\text{rtg}}\right)$

46:          $\bar{y}_{t_{k+h}}^{(b)} \leftarrow \frac{1}{N}\sum_{n=1}^{N}\hat{y}_{t_{k+h}}^{(b,n)}; \quad \hat{o}_{t_{k+h}}^{(b)} \leftarrow \text{Obs}(\bar{y}_{t_{k+h}}^{(b)})$

47:        **end for**

48:        Select top-$\kappa_{\text{keep}}$ from $\{K_{t_{k+h-1}}^{(b)}\}_{b=1}^{B}$ by score and attach as children $(\hat{o}_{t_{k+h}}^{(b)}, K_{t_{k+h-1}}^{(b)})$ under $(\hat{o}, K^c)$ in $\mathcal{T}$

49:      **end for**

50: **end for**

51: **return** $\mathcal{T}$

## A.2 DECISION TIME VS CONTEXT LENGTH

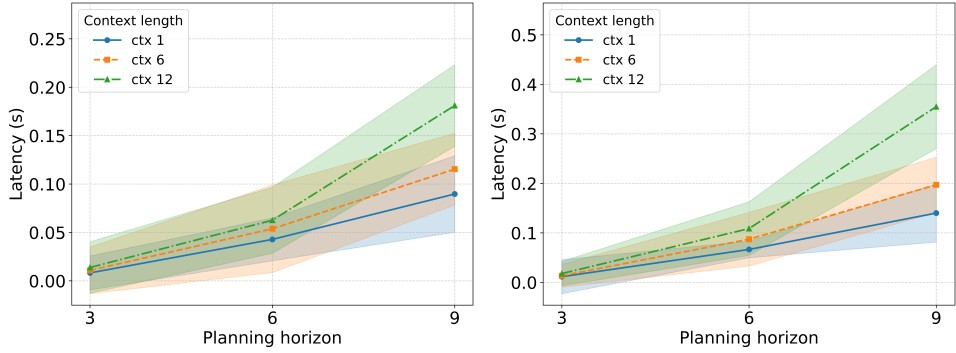

Figure 6: Average Decision Time vs. Context Length for initial number of samples of 16 (left) and 32 (right).

The latency results clearly highlight the trade-off between context size, planning depth, and decision-making efficiency. Our analysis of decision latency versus context length reveals that longer contexts substantially increase the average decision time, particularly as the planning depth increases. For instance, at planning horizon 9, decision latency escalates sharply from approximately 0.09 seconds (context size 1) to 0.35 seconds (context size 12) when the initial number of samples is 32. Similarly,

at lower planning depths (1 and 2), longer contexts still cause noticeable latency increases, albeit less dramatically.

Thus, in scenarios characterized by dynamics influenced by unobserved states, our findings underline the necessity of carefully selecting context lengths. It is crucial to encode historical observations effectively, extracting the most informative signals without redundant details that contribute minimal additional predictive power but significantly raise computational latency. In practical applications, particularly those involving stochasticity and continuous state-action spaces, balancing sufficient context length to maintain accuracy against the latency imposed by deeper planning becomes a critical design decision. Efficient decision-making, therefore, relies on identifying a context size that captures the essential dynamics without incurring unnecessary computational overhead.

### A.3 EXPERIMENT DETAILS

#### A.3.1 IMPLEMENTATION DETAILS

The hyper-parameters setting of I-TAP is presented in Table 3. For baselines, we adopt similar hyper-parameters settings as suggested by original works, including TAP (Jiang et al., 2023), L-MAP (Luo et al., 2025), CQL (Kumar et al., 2020), IQL Kostrikov et al. (2022), DT (Chen et al., 2021), 1R2R (Rigter et al., 2023). Each run of I-TAP takes approximately 6 hours on 1 NVIDIA RTX5090 GPU and Intel(R) Core(TM) i9-14900KS.

#### A.3.2 DOMAINS

**Stochastic MuJoCo.** The Stochastic MuJoCo tasks introduced by Rigter et al. (2023) apply incremental perturbation forces following a uniform random walk (Popko et al., 2016). At each timestep, the perturbation force $f_t$ is updated as:

$$f_{t+1} = f_t + \Delta f, \quad \Delta f \sim \text{Uniform}\left(-0.1 \cdot f_{\text{MAX}}, \ 0.1 \cdot f_{\text{MAX}}\right), \tag{1}$$

with the total perturbation clipped to remain within $[-f_{\text{MAX}}, f_{\text{MAX}}]$. This model introduces persistent, incremental perturbations, representing a baseline scenario. For both Hopper and Walker2D environments, the perturbation force $f_t$ is applied horizontally along the x-axis to simulate external disturbances, such as wind gusts. The maximum perturbation magnitude $f_{\text{MAX}}$ specifically for the Hopper moderate perturbation level is 2.5 Newtons, high perturbation level is 5 Newtons, Walker2D moderate perturbation level is 7 Newtons, high perturbation level is 12 Newtons.

**Adroit as a POMDP.** Many Adroit manipulation tasks expose *privileged* channels that directly encode the goal state and progress (e.g., target poses, object–target deltas, insertion depth). To evaluate agents under partial observability, we define a POMDP variant that preserves the original dynamics and reward but *ablates* these channels at evaluation time. Concretely, let the environment produce an observation $o_t \in \mathbb{R}^d$ under the standard `v1` layout. We introduce a fixed binary mask $m \in \{0, 1\}^d$ (zeros at privileged indices) and set

$$\tilde{o}_t \ = \ m \odot o_t, \qquad O(\tilde{o}_t \mid s_t) \ = \ \delta\big(\tilde{o}_t - m \odot o_t\big),$$

yielding a POMDP $\big(\mathcal{S}, \mathcal{A}, \mathcal{T}, r, \gamma, \Omega, O\big)$ in which progress toward the goal must be *inferred* from history rather than read off directly.

**Masking regimes.** Indices refer to 0-based positions in the default `v1` observation vector.

- **Pen** ($d{=}45$): $\mathcal{I} = \{36, 37, 39, 40, 42, 43\}$.
- **Relocate** ($d{=}39$): $\mathcal{I} = \{30, \dots, 38\}$.
- **Door** ($d{=}39$): $\mathcal{I} = \{27, 28, 32, \dots, 38\}$.
- **Hammer** ($d{=}46$): $\mathcal{I} = \{43, 44, 45\}$.

This construction leaves proprioception and contact signals intact while removing privileged goal vectors and progress proxies, thereby converting the original fully observable tasks into history-dependent POMDPs that better reflect realistic sensing and require temporal credit assignment and state estimation.

Table 3: List of Hyper-parameters

| Environment | Hyper-parameter | Value |
|---|---|---|
| All | learning rate | $1 \times 10^{-4}$ |
| All | batch size | 512 |
| All | dropout probability | 0.1 |
| All | number of attention heads | 4 |
| All | macro action length $L$ | 3 |
| All | embedding size (latent code) | 512 |
| All | $c_1$ | 1.25 |
| All | $c_2$ | 19652 |
| MuJoco | context length $c$/training sequence length | 6,3 |
| MuJoco | discount factor | 0.99 |
| MuJoco | number of Transformer layers | 4 |
| MuJoco | feature vector size | 512 |
| MuJoco | codebook size | 512 |
| MuJoco | initial number of policy samples $M$ | 16 |
| MuJoco | number of transition samples $N$ | 4 |
| MuJoco | number of policy samples $B$ | 4 |
| MuJoco | number of MCTS iterations | 100 |
| Mujoco | $\kappa_{\text{keep}}$ | 50% |
| MuJoco | temperature | 2 |
| MuJoco | Residual Depth | 1 |
| MuJoco | planning horizon | 9 |
| Adroit | context length $c$/training sequence length | 6/24 |
| Adroit | discount factor | 0.99 |
| Adroit | number Transformer layers | 4 |
| Adroit | feature vector size | 256 |
| Adroit | codebook size | 512 |
| Adroit | initial number of policy samples $M$ | 16 |
| Adroit | number of transition samples $N$ | 4 |
| Adroit | number of policy samples $B$ | 4 |
| Adroit | number of MCTS iterations | 100 |
| Adroit | Residual Depth | 2,3 |
| Adroit | $\kappa_{\text{keep}}$ | 10% |
| Adroit | $J$ | 4 |
| Adroit | temperature | 1 |
| Adroit | planning horizon | 3 |

## B    STATEMENT ON LLM USAGE

We used a large language model solely for light copy-editing of this manuscript (grammar, phrasing, and stylistic polishing). The LLM did not contribute to research ideation, problem formulation, algorithm design, experiments, analysis, or the creation of technical content. All methods, results, and citations were authored and verified by the authors, and any LLM-suggested wording was reviewed and edited for accuracy. No references were generated by the LLM. LLMs are not authors.

