# OpenReview forum: "In‑Context Planning with Latent Temporal Abstractions"
_ICLR.cc/2026/Conference — ICLR 2026 Conference Withdrawn Submission_

### Official Review · Reviewer_UDdL · 2025-10-22

**Soundness:** 3
**Presentation:** 2
**Contribution:** 1
**Rating:** 2
**Confidence:** 3

**Summary:**

The paper proposes I-TAP, an offline RL framework that learns a residual-quantized VAE over observation–macro-action sequences plus a depth-aware temporal Transformer, and then plans online with MCTS directly in the latent token space. The motivation is to handle the latent variables and unify in-context learning and online planning. The goal is efficient in-context adaptation under stochastic dynamics and partial observability, with experiments on (stochastic) MuJoCo locomotion and Adroit manipulation.  The performance improvement is impressive.

**Strengths:**

1.	Clear motivation – Nicely frames in-context planning with temporal abstraction as a way to cut down horizon and branching while adapting on the fly under partial observability.
2.	Method coherence – The combination of a residual-quantized VAE, temporal Transformer prior, and MCTS over latent tokens is technically sound and well integrated.
3.	Strong empirical showing – I-TAP performs consistently well across stochastic MuJoCo and Adroit, even under partial observability, often outperforming strong baselines.

Besides these points, I believe modeling latent space with contextual adaptation is a principled direction for RL planning. It efficiently captures latent confounders and updates them through dynamics, making this line of work fundamental and heuristic for many downstream tasks.

**Weaknesses:**

I am mainly concerned about its novelty and presentation, see below:

**Novelty (or how this paper convey it)**:

1. The method closely follows L-MAP/TAP (discrete temporal abstraction + MCTS). Residual quantization and context-conditioning feel like engineering tweaks rather than a conceptual leap. The paper doesn’t clearly justify why these changes matter or how they lead to better performance.

2. Improvements seem mostly empirical; there’s no ablation or clear rationale for each design choice (e.g., residual quantization, context use, or planning depth). Hard to tell which part drives the gains.

3. Relies on short context windows instead of a belief-state or recurrent model. It’s unclear whether improvements come from longer context, deeper planning, or the RQ prior.

4. Baselines like DT can be sensitive to tuning, context length, and stochasticity. The paper doesn’t confirm consistent capacity or fair tuning across methods, so some reported advantages may not hold.

5. The paper does not explicitly evaluate in-context adaptive planning under true OOD conditions—the core motivation of such methods (e.g., [1]). Although verified rewards adapt the latent space, this occurs only within the training distribution, making the claimed adaptation and generalization unclear. If reusing the in-distribution contextual samples, where is the adaptation?

**Presentation**:

1. Figure 1 doesn’t clearly convey the pipeline, and Figure 2 is cluttered and hard to read. A simpler schematic or step-by-step breakdown would help.

2. Important details like MCTS budget, top-K truncation, and temperature schedules are scattered. Should organize them well for readability.

Generally speaking, this paper could be shown in a more principled way, but currently it falls below the ICLR standard.

[1] Xu, Mengdi, et al. "Prompting decision transformer for few-shot policy generalization." ICML 2022.

**Questions:**

1. Can you compare to a belief-state or recurrent baseline to isolate the benefit of context-based planning vs. explicit state estimation?

2. What are the codebook sizes and residual depths used per domain? How sensitive is performance to these choices?

3. Could you carefully clarify the difference between this work and [2] and the motivation for making such a difference? If not, it appears to be an incremental research.

[2] Luo, Baiting, et al. "Scalable Decision-Making in Stochastic Environments through Learned Temporal Abstraction." arXiv preprint arXiv:2502.21186 (2025).

---

### Official Review · Reviewer_Avwq · 2025-10-29

**Soundness:** 2
**Presentation:** 3
**Contribution:** 2
**Rating:** 4
**Confidence:** 3

**Summary:**

The manuscript proposes I-TAP, a framework that unifies in-context adaptation and planning within a learned latent temporal-abstraction latent space. The framework constructs temporal abstractions by discretizing state-macro-action sequences into residual tokens, coupled with a residual temporal transformer prior to generate future sequences based on past experiences in the latent token space. The learned prior will subsequently be used for planning during inference, combined with P-UCT for targeted exploration, supported by in-distribution behaviors. The framework achieves strong performance on MuJoco and Adroit tasks under stochastic noises and partial observability, and remains robust with suboptimal behavioral policies.

**Strengths:**

- I-TAP integrates in-context adaptation and planning on a latent temporal abstraction space through sequence modeling and residual quantization, effectively utilizing historical experience and future planning for informative decision making at an abstracted temporal scale
- Planning over macro-action space helps decision latency
- The method is shown to be robust when handling stochastic dynamics, partial observability in continuous control tasks with behavior policies of varying quality

**Weaknesses:**

- As operating on temporal abstractions mainly aims to facilitate long-horizon decision making, it would be crucial to evaluate the proposed method on long-horizon/sparse-reward tasks, for example, AntMaze-Medium/Large, which ask the agent to navigate through a long path to reach the goal position
- Trajectory Transformer is an important baseline that performs planning as beam search on sequence modeling, which should be included in Table 1 and 2
- To fully understand the in-context adaptation capabilities of the proposed method, I wonder if it's possible to dynamically change the hidden parameter (i.e., increase the noise in MuJoCo) throughout the rollout of one episode and see if the model can adjust to the environmental changes robustly

**Questions:**

- How many rollouts and seeds are used to calculate the aggregated results?

---

### Official Review · Reviewer_2AZw · 2025-10-30

**Soundness:** 2
**Presentation:** 1
**Contribution:** 1
**Rating:** 2
**Confidence:** 4

**Summary:**

The paper proposes to employ the residual quantization VAE (RQ-VAE) to replace the VQ-VAE in L-MAP which uses VQ-VAE and learns a state-macro action dynamics model.

**Strengths:**

- Learning a proper dynamics model for planning is important;

**Weaknesses:**

- The contribution using RQ-VAE to replace VQ-VAE is incremental;
- There are many unclear concepts used without proper explanation and readers need to refer to the L-MAP paper to understand them. The paper seems highly relying on the L-MAP paper.
- Many claims are not properly justified. See the questions.

**Questions:**

- What is the difference between $\mathcal{G}^{(L)}$ and $\tilde{r}(s_t, m, w)$;
- Both *in context RL* and *temporal abstraction* should be more explicitly defined;
- Line 158-161 said that “prior work based on state-conditioned VQ-VAE suffers from high dimensionality, while the proposed observation-conditioned RQ-VAE improves scalability”, which looks strange to me. Usually, the observation space (e.g., camera images) is much higher than the underlying state space (e.g., the abstract state information). Maybe you should explain more or correct that sentence.
- In line 189-190, *residual depths* is used without proper explanation. Although Lee et al. 2022 is referred, as a critical concept, it should be explained in a standalone way.
- In line 255-258, how is the in-context planning implemented other than using MCTS as a planner? Furthermore, by taking the expectations over futures, how MCTS decouples action selection from noisy return estimates?

---

### Note · Authors · 2025-11-12

**Comment:**

We sincerely appreciate the efforts of the AC and reviewers and have decided to withdraw the paper so that we can further improve the draft.

**Withdrawal Confirmation:**

I have read and agree with the venue's withdrawal policy on behalf of myself and my co-authors.